# SIMKEY: A SEMANTICALLY AWARE KEY MODULE FOR WATERMARKING LANGUAGE MODELS

## ABSTRACT

The rapid spread of text generated by large language models (LLMs) makes it increasingly difficult to distinguish authentic human writing from machine output. Watermarking offers a promising solution: model owners can embed an imperceptible signal into generated text, marking its origin. Most leading approaches seed an LLM's next-token sampling with a pseudo-random key that can later be recovered to identify the text as machine-generated, while only minimally altering the model's output distribution. However, these methods suffer from two related issues: (i) watermarks are brittle to simple surface-level edits such as paraphrasing or reordering; and (ii) adversaries can append unrelated, potentially harmful text that inherits the watermark, risking reputational damage to model owners. To address these issues, we introduce SIMKEY[1], a semantic key module that strengthens watermark robustness by tying key generation to the *meaning* of prior context. SIMKEY uses locality-sensitive hashing over semantic embeddings to ensure that paraphrased text yields the same watermark key, while unrelated or semantically shifted text produces a different one. Integrated with state-of-the-art watermarking schemes, SIMKEY improves watermark robustness to paraphrasing and translation while preventing harmful content from false attribution, establishing semantic-aware keying as a practical and extensible watermarking direction.

## 1 INTRODUCTION

As large language models (LLMs) become widely deployed across various domains, concerns regarding the authenticity and provenance of AI-generated text have grown significantly (Bian et al., 2024; Hanley & Durumeric, 2024; Pan et al., 2023). Watermarking techniques offer a crucial mechanism for distinguishing between human-authored and machine-generated content (Kuditipudi et al., 2024; Yang et al., 2023). Ideally, a watermarking method should not only provide reliable identification of AI-generated text but also maintain high generation quality. To be practical, watermarks must also be robust against adversarial attempts to remove the watermark or to mark unrelated content.

These practical considerations make embedding watermarks into generated text inherently challenging. Most methods use a *mark module* that modifies token generation, and a *key module* that conditions the mark module on previously generated text Huang & Wan (2024) (see Figure 1). Early approaches use a mark module that increases the likelihood of certain token sequences (e.g. the now canonical "red-green" list (Zhao et al., 2023a; Kirchenbauer et al., 2024)), but this often introduces fluency-degrading distortions (Rastogi & Pruthi, 2024). Moreover, such patterns can be exploited by adversaries (Sadasivan et al., 2023; Jovanović et al., 2024). To mitigate these issues, other methods leverage *pseudo-random next-token selection*. Concretely, they use a secret random variable (i.e. the *key*) to seed the sampling process, which keeps outputs consistent with the LLM distribution (Sadasivan et al., 2023; Kuditipudi et al., 2024; Liu et al., 2025).

While watermark *embedding* techniques have been widely studied, *key* generation remains largely unchanged. Keys can be generated through a context-*independent* key module; e.g., by using a cyclic key or sampling from a given key pool. However, reusing keys introduces patterns that undermine both the quality and security of the generated text (Kuditipudi et al., 2024). Using many different keys reduces these effects, but yields watermarks that are harder to detect and computationally less efficient. A context-*dependent* approach might hash prior tokens to generate keys; however,

---

[1]The full code can be found at XXXXX

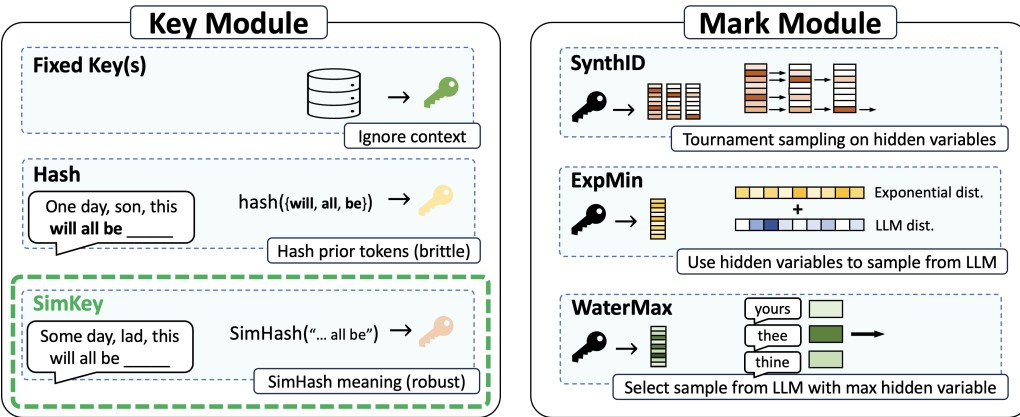

Figure 1: **Common components in watermarking.** The *key module* (left) generates a seed that guides watermarking, using options such as a fixed key (or a fixed set of keys), a hash of prior tokens, or a semantic SimHash of the context (ours). The *mark module* (right) modifies token sampling given the key. E.g., via tournament sampling (SynthID), exponential-min sampling (ExpMin), or selecting generations that maximize a hidden variable (WaterMax).

hashing the last few tokens causes the model to reuse common phrases, and introduce brittleness to small context changes (Kirchenbauer et al., 2024). Additionally, both context-*independent* and -*dependent* approaches risk compromising the watermark owner's reputation, since an adversary can insert harmful text into a watermarked passage, which will still be flagged as model-generated.

To address this, we introduce a key module that uses the locality-sensitive hashing (LSH) technique SimHash (Charikar, 2002). By applying SimHash to a semantic embedding of the preceding context, our method ties the key to the *meaning* of the text. As a key module, SIMKEY is **(i) robust to semantic paraphrasing**: when meaning is preserved (i.e., semantic embeddings are similar), the key tends to remain the same. Simultaneously, SIMKEY is **(ii) sensitive to meaning-changing edits**: if watermarked text is moved out of context or if unrelated (potentially harmful) tokens are inserted, the key is likely to change. Finally, SIMKEY ensures **(iii) a sufficiently large and diverse key space**, since SIMKEY varies the key with semantics and across multiple hash identities.

We emphasize that SIMKEY is a general key module that can be paired with many different mark modules. In this work, we demonstrate SIMKEY combined with three state-of-the-art mark modules: distortion-free exponential minimum sampling (ExpMin) (Kuditipudi et al., 2024), SynthID (Dathathri et al., 2024), and WaterMax (Giboulot & Furon, 2024). We describe how to use SIMKEY in Section 3. In Section 4, we evaluate SIMKEY and confirm it is sensitive to unrelated (potentially harmful) content insertion while remaining robust to meaning-preserving transformations such as paraphrasing and translation. We end with a discussion of limitations and broader implications.

## 2 PRELIMINARIES

**SimHash**

Originally developed for efficient approximate nearest neighbor search, locality sensitive hashing (LSH) provides embeddings that preserve similarity (Indyk & Motwani, 1998; Gionis et al., 1999). SimHash (Charikar, 2002) is one such LSH approach that embeds an input vector by random projections so that similar inputs yield similar bit patterns. The benefit of SimHash over standard hashing is that nearby vectors $\mathbf{v}$ and $\mathbf{v}'$ are more likely to agree on bits, with agreement controlled by the angle between them:

$$\theta(\mathbf{v}, \mathbf{v}') = \arccos(\frac{\langle \mathbf{v}, \mathbf{v}' \rangle}{\|\mathbf{v}\|_2 \|\mathbf{v}'\|_2}). \quad (1)$$

To implement SimHash, we choose $b$ random unit vectors $\{\mathbf{r}_j\}_{j=1}^b$, project the vector $\mathbf{v}$ onto each, and record the sign to produce a $b$-bit sequence, which we then hash to obtain a pseudo-random

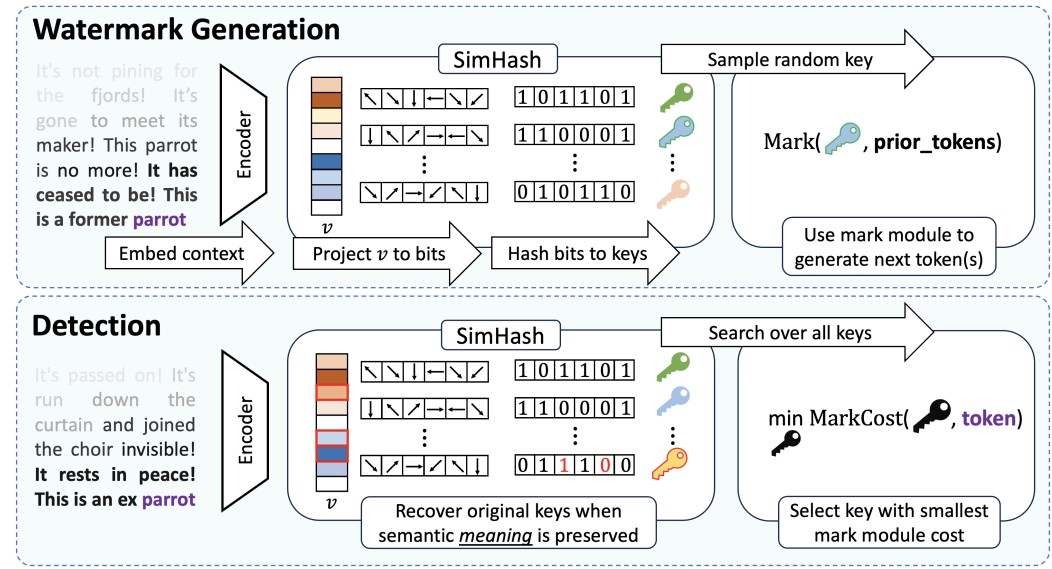

Figure 2: **Overview of our semantic watermarking method, SIMKEY.** *Generation (top):* we embed the preceding context into a semantic vector $v$, project onto random directions, and take signs (SimHash), then hash the resulting bits to seed keys that modulate the LLM sampling (e.g., Gumbel/ExpMin sampling). *Detection (bottom):* we re-embed the context before each token, recompute SIMKEY, and use the mark module's alignment cost to select the best-matching key per position.

output (Algorithm 1). The probability of reproducing the same key for two semantic embeddings $\mathbf{v}$ and $\mathbf{v}'$ is then given as a function of the angle $\theta(\mathbf{v}, \mathbf{v}')$ in Lemma 2.1.

**Lemma 2.1** (SimHash Guarantee (Charikar, 2002))**.** *Consider two vectors $\mathbf{v}$ and $\mathbf{v}'$, with angle $\theta(\mathbf{v}, \mathbf{v}')$. For a fixed input (i.e., the same secret salt and key index), Algorithm 1 produces the same key with probability ,* $\left(1 - \frac{\theta(v,v')}{180°}\right)^b$.

Increasing $b$ will decrease the probability of a match in the key, especially for far apart vectors with low angle.

**Mark Modules** The *mark module* is the part of any existing watermarking technique that modifies the next token generation of the underlying LLM. SIMKEY is flexible and compatible with most existing mark modules. To apply SIMKEY to existing methods, we need only assume their mark module provides two functions:

- Mark: maps a random key (provided by SIMKEY) and the prior tokens to the next token(s), using an LLM, and any internal watermarking logic.

---

**Algorithm 1** SIMKEY

**Input:** $\mathbf{v}$: semantic vector, idx: key index, salt: secret salt, $b$: number of bits, hash: cryptographic hash function
**Output:** Semantically and securely generated key
bits $\leftarrow \mathbf{0}$              ▷ Initialize hash input
**for** $j = 1, \ldots, b$ **do**
  $s \leftarrow \text{hash}(\text{idx}, j, \text{salt})$
  Sample $\mathbf{r}_j \overset{s}{\sim} \mathcal{N}(\mathbf{0}, \mathbf{I})$    ▷ Reproducibly sample random projection vector *with* key index
  bits$[j] \leftarrow \text{sign}(\langle \mathbf{v}, \mathbf{r}_j \rangle)$        ▷ Random projection
**end for**
key $\leftarrow \text{hash}(\text{bits}, \text{idx}, \text{salt})$
**return** key

---

---

**Algorithm 2** Generation with SIMKEY

---

**Input:** `Mark`: mark module for generating next tokens from a key, `tokens`: prior tokens, $V$: vocabulary size, $k$: number of used hash function identities, $b$: number of bits, `salt`: secret salt
**Output:** Watermarked token drawn from LLM distribution
$\mathbf{v} \leftarrow \text{Embed}(\texttt{tokens})$          ▷ Semantically embed prior tokens
$\text{idx} \sim \text{Uniform}(\{1, \ldots, k\})$          ▷ Randomly select key index
$\text{key} \leftarrow \text{SIMKEY}(\mathbf{v}, \text{idx}, \texttt{salt})$ (i.e. Algorithm 1)
$\texttt{next\_token} \leftarrow \text{Mark}(\text{key}, \texttt{tokens})$    ▷ Sample next token using watermark method and key
**return** `next_token`

---

- `MarkCost`: maps a key and a token to an *alignment cost*, a real number measuring the likelihood that the generated token was produced by `Mark`, given a candidate key. Without loss of generality, we assume a lower cost indicates a higher likelihood.

## 3 SIMKEY- SEMANTIC AND DISTORTION FREE WATERMARKING

Our goal is to attach a watermark to the *meaning* of text rather than to *exact* token sequences. This serves two purposes: we want watermarks to *persist* when text is paraphrased to obscure origin, *and* we would like the watermark to *disappear* if unrelated, potentially harmful content is added.

Existing watermarking methods often fail to achieve this because their detection depends on the exact sequence of preceding tokens rather than their meaning. This makes them vulnerable to removal attacks, where even minor rewordings can erase the watermark. Conversely, SIMKEY, computes a semantic embedding of prior context and applies SimHash to produce the key. Importantly, SIMKEY is not itself a new watermarking scheme. Instead, it is a general and flexible component that can augment existing schemes, improving their robustness. To demonstrate how it works with existing `Mark` modules, we integrate SIMKEY into (1) distortion-free exponential minimum sampling (Exp-Min) (Kuditipudi et al., 2024), (2) tournament-style sampling (SynthID) (Dathathri et al., 2024), and (3) max-normal style sampling (WaterMax) Giboulot & Furon (2024). We describe the creation and detection procedures in the following subsections and provide pseudocode in Algorithms 2 and 3.

### 3.1 KEY GENERATION

Our goal with SIMKEY is to attach a key to *what the context means*, not merely to *what the last few tokens were*. Concretely, before each generation step, SIMKEY embeds the prior context into a semantic vector $\mathbf{v}$ that captures its meaning[2]. SIMKEY then applies SimHash (Equation (1)) to convert that vector into a compact, reproducible bit pattern: we project $\mathbf{v}$ onto $b$ random directions and encode the signs of the projections as bits:

$$\texttt{bits} = \left[\text{sign}(r_1^\top v), \text{sign}(r_2^\top v), \ldots, \text{sign}(r_b^\top v)\right] \in \{-1, 1\}^b \,,$$

Next, we use a cryptographic hash to produce the `key`, which we pass to `Mark` module to sample the next token from the underlying watermarking scheme. The full procedure is given in Algorithm 1

**Key index variation.** In long generations the same semantic state can reappear (following similar context embeddings), which risks reusing identical keys too often. We therefore randomly draw an index `idx` from $\{1, \ldots, k\}$ at each step, effectively selecting among $k$ independent SimHash instances within SIMKEY. This maintains semantic stability, in that as long as the meaning remains similar, the right key can still be recovered for *some* index. At the same time it reduces repetitive key reuse that could harm fluency.

### 3.2 WATERMARK DETECTION

During the detection phase, our goal is to recover the same key in order to determine whether the text was likely generated using the watermark. To recover the key, for each position $i$, we re-embed the

---

[2]Recent advances in language modeling have made powerful semantic embedders abundant; in this paper we use `all-MiniLM-L6-v2` (Reimers & Gurevych, 2019), a sentence-transformer model that encodes input text into 384-dimensional embeddings, although many other similar models are available.

---

**Algorithm 3** SIMKEY Detection

---

**Input:** `tokens`: (possibly) watermarked tokens, `MarkCost`: a mark module specific function for computing the alignment cost between the tokens and a key, $V$: vocabulary size, $k$: number of keys, $b$: number of bits, `salt`: secret salt
**Output:** $p$-value: Probability of observing tokens if they were *not* watermarked
$\text{cost} \leftarrow 0$
**for** $i = 1, \ldots, |\texttt{tokens}|$ **do**
  $\texttt{prior\_tokens} \leftarrow \{\texttt{tokens}_1, \ldots, \texttt{tokens}_{i-1}\})$
  $\mathbf{v} \leftarrow \text{Embed}(\texttt{prior\_tokens})$            ▷ Semantically embed prior tokens
  $\text{cost}_i \leftarrow \infty$
  ▷ Check each key index
  **for** $\texttt{idx} = 1, \ldots, k$ **do**
    $\text{key} \leftarrow \text{SIMKEY}(\mathbf{v}, \texttt{idx}, \texttt{salt})$          ▷ (i.e. Algorithm 1)
    $\text{cand\_cost}_{\texttt{idx}} \leftarrow \text{MarkCost}(\text{key}, \texttt{prior\_tokens})$ ▷ Candidate cost of instance idx
    $\text{cost}_i \leftarrow \min(\text{cost}_i, \text{cand\_cost}_{\texttt{idx}})$
  **end for**
  $\text{cost} \leftarrow \text{cost} + \text{cost}_i$
**end for**
$p\text{-value} \leftarrow \Pr(\text{observing a value as large as cost})$     ▷ Depends on alignment cost distribution
**return** $p$-value

---

preceding context to obtain $\mathbf{v}'$ and run SIMKEY across all $k$ indices to reconstruct candidate keys. Because the text may have been manipulated between generation and detection, $\mathbf{v}'$ may differ from the original context embedding $\mathbf{v}$. Nonetheless, because SimHash depends on *semantic* similarity rather than exact token identity, edits that largely preserve meaning are likely to yield the same key during detection as during generation (the formal probability guarantee is given by Lemma 2.1).

Still, even if the watermarked text is unchanged, we do not know which key index was used during generation. To address this, we evaluate all $k$ candidate key indices, where each one is associated with an *alignment cost* noted as $\text{cand\_cost}_{\texttt{idx}}$, and select the one yielding the minimum cost (as defined by the mark module). The per-token minimum costs are then summed across the entire sequence. We formally describe the procedure in Algorithm 3.

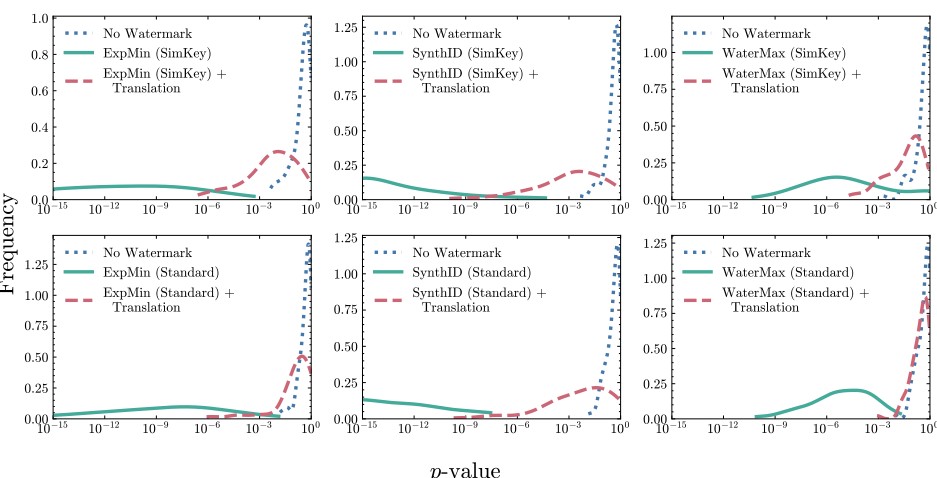

Figure 3: **Per-token watermark detectability with SIMKEY and standard hashing**. The $p$-value distributions by mark module (columns) and key module (rows). Shown from $10^0$ to $10^{-15}$, values below this are truncated. SIMKEY gives similar detectability to standard hashing on the original text. However, after watermarked text is translated to a second language and back, SIMKEY is more robust, since the key depends on the *meaning* of the text rather than a set of precise tokens.

Finally, to assess whether the resulting total cost provides sufficient evidence of watermarking, we compute a $p$-value: **the probability of observing a cost at least this low under the null hypothesis** that the text was *not* generated with a watermarking scheme. The formal calculation is given below.

### $p$-value Computation

Let $\mathcal{D}$ be the distribution of the alignment cost when `key` was *not* used to generate the tokens. We assume here that $\mathcal{D}$ is a discrete distribution. Let $F : \text{supp}(\mathcal{D}) \to [0, 1]$ be the CDF of $\mathcal{D}$. The CDF of the minimum of $k$ independent draws from $\mathcal{D}$ is given by:

$$1 - F_{\text{cost}}^1(y) = \Pr(\min_{\text{idx} \in \{1,\ldots,k\}} \texttt{cand\_cost}_{\text{idx}} > y) = \prod_{\text{idx}=1}^{k} (1 - F(y)) = (1 - F(y))^k \quad (2)$$

where the second equality follows by independence.

We now define the CDF of the sum of $\ell$ independent $\texttt{cost}_i$ samples. For $\ell > 1$, we recursively define:

$$F_{\text{cost}}^\ell(y) = \sum_{z \in \text{supp}(\mathcal{D})} F_{\text{cost}}^{\ell-1}(z) F_{\text{cost}}^1(y - z) = \text{Convolve}(F_{\text{cost}}^{\ell-1}, F_{\text{cost}}^1) . \quad (3)$$

Finally, we can compute the $p$-value via

$$F_{\text{cost}}^{|\texttt{tokens}|}(\texttt{cost}) . \quad (4)$$

When the alignment-cost distribution is continuous, the $p$-value can be computed either by discretizing the support or, in some cases, in closed form. For instance, under ExpMin, the alignment cost follows an exponential distribution, yielding a closed form $p$-value (see Appendix A.1 for details).

**Inheriting distortion-free properties**  SIMKEY only replaces the key-generation mechanism while leaving the mark module and the underlying LLM logits unchanged. As a result, when SIMKEY is paired with a distortion-free mark module such as ExpMin (Kuditipudi et al., 2024), the overall watermarking scheme may remain distortion-free in the sense of preserving the base model's marginal next-token distribution: SIMKEY alters only the correlation between generated tokens and the hidden randomness used by the mark module, not the distribution over tokens itself.

## 4  RESULTS

**Experimental Setup.**  We conduct all experiments using HuggingFace's `transformers` library and implement watermarking methods through the `LogitsProcessor` interface. For text generation, we use top-$p$ sampling with $p = 0.9$ to maintain comparable diversity across methods.

***Base Model.***  For the experiments reported in the main text, we use the quantized version of the Meta Llama 3.1 instruction-tuned 70B model `hugging-quants/Meta-Llama-3.1-70B-Instruct-AWQ-INT4` and the Meta Llama 3 8B model `meta-llama/Meta-Llama-3-8B`. We employ the 70B model for Figure 5, and Table 1, and the 8B model for Figures 3 and 4 due to computational constraints.

***Prompt Initialization.***  Prompts are sampled by drawing three random words to form a short phrase, which serves as a neutral starting point for generation. This procedure avoids strong topical bias while ensuring syntactically valid completions.

***Watermarking Parameters.***  For a fair comparison across all methods, we set the number of keys $k = 4$ and the number of bits $b = 4$ for the parameters of SIMKEY. We also set the context window as 8 tokens for all methods and key modules. Unless otherwise noted (e.g., in our long-context, hyperparameter, and multilingual robustness experiments), we use this $(b, k) = (4, 4)$ configuration.

***Perturbations and Attacks.***  To evaluate robustness, we apply several classes of meaning-preserving and meaning-altering perturbations, which are standard attacks from the watermarking literature (Liu et al., 2024b). The first such attack is the **Translation Attack**, which involves translating the generated text from English to French and then back to English (we use the `opus-mt-tc-big-en-fr` and `opus-mt-tc-big-fr-en` translation models (Junczys-

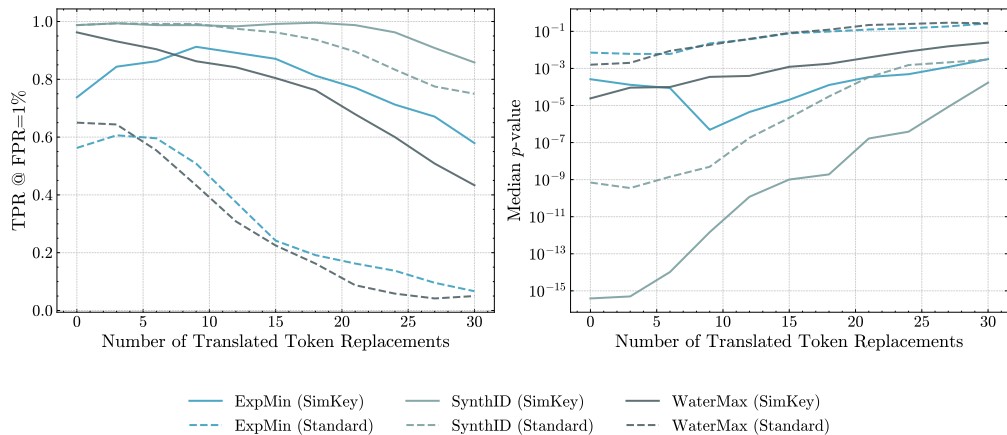

Figure 4: **SIMKEY substantially improves detectability under translated token replacements.** At a fixed false positive rate of 1%, SIMKEY substantially improves the true positive rate for ExpMin and WaterMax. At first glance, it may appear that SimKey is less effective for SynthID; however, SimKey actually reduces the median $p$-value of SynthID by *several orders of magnitude*.

Dowmunt et al., 2018)). Translation preserves the overall semantic meaning of the text while perturbing the surface form, sometimes drastically. We also apply **Translated Token Substitution**, randomly translating individual tokens to French and back, which yields subtle lexical variations without changing either sentence length or global meaning.

For the **Unrelated Token Substitutions** attack, we randomly replace selected token positions with uniformly sampled vocabulary IDs, preserving sequence length but disrupting the semantic meaning. Conversely, for the **Related Token Substitution** perturbation, we randomly mask tokens and use the BERT base model (cased) `google-bert/bert-base-cased` to choose the most probable replacement. We repeat this process until a replacement different from the original token is obtained, ensuring that the substituted word remains contextually plausible while subtly altering form.

To test whether semantic keying remains effective in **Longer Generations**, we ran an additional experiment with 500-token outputs and applied the same *unrelated* and *related* token substitution attacks as in Table 1, now replacing 50 tokens per sample (Table 2). We also evaluated a common perturbation (*not* an explicit attack): a **Conversational Setting** in which an unwatermarked LLM poses a scripted sequence of three prompts forming a coherent dialogue, while a SIMKEY-watermarked model (using ExpMin as the mark module) produces short responses of 10-30 words. The topics were chosen so that the conversation remains coherent but gradually shifts in focus (e.g., from shoe recommendations to training plans, or from plant care to interior design). Only the model's replies are watermarked; the user prompts are fixed and unwatermarked.

Finally, to approximate a stronger and more adaptive adversary, we also evaluate an **LLM-in-the-loop paraphrasing attack** inspired by prior work on watermark removal (Zhang et al., 2024a). Starting from watermarked text (using ExpMin or WaterMax as the mark module and either standard hashing or SIMKEY as the key module), we apply a separate paraphraser LLM (`GPT 5.1 inference`) twice in succession, instructing it to rewrite the passage while preserving meaning.

All experimental details (cost, compute, prompts, etc.) are detailed in Appendix D.

**Summary.** We evaluate SIMKEY as a key module paired with three mark modules: ExpMin (Kuditipudi et al., 2024), SynthID (Dathathri et al., 2024), and WaterMax (Giboulot & Furon, 2024), with a standard (non-semantic) hashing key as the baseline. Here, we briefly summarize our results before offering a closer analysis: **(1)** On clean text, SIMKEY matches the $p$-value distribution of standard hashing (Fig. 3); **(2)** Under meaning-preserving edits (paraphrase/translation), SIMKEY substantially improves detectability (Fig. 4, Table 1); **(3)** Under meaning-changing edits (unrelated insertions/replacements), SIMKEY degrades similarly to standard hashing (Table 1), as desired; and **(4)** Perplexity/distortion is essentially unchanged with respect to standard hashing, with differences dominated by the choice of mark module (Fig. 5).

Table 1: **True Positive Rate at Fixed False Positive Rate under different transformations (TPR@FPR≤1%).** For each method we perturb the text by changing tokens to related or unrelated tokens (under two settings, 15 and 30 modifications). We examine each attack with the standard hashing method (St. Hash) or our method (SIMKEY). Across both 15 and 30 token replacements we find that SIMKEY is more robust to related token replacements while being similarly sensitive to the baseline standard hashing scheme to unrelated token replacements.

| Method | Unrelated Attack | | | | Related Attack | | | |
| | 15 Tokens | | 30 Tokens | | 15 Tokens | | 30 Tokens | |
| | St. Hash | SimKey | St. Hash | SimKey | St. Hash | SimKey | St. Hash | SimKey |
|---|---|---|---|---|---|---|---|---|
| ExpMin | 0.063 | **0.075** | 0.013 | **0.038** | 0.025 | **0.450** | 0.063 | **0.325** |
| SynthID | **0.813** | 0.700 | **0.225** | 0.138 | 0.500 | **0.875** | 0.288 | **0.800** |
| WaterMax | 0.075 | **0.250** | 0.013 | **0.050** | 0.025 | **0.613** | 0.013 | **0.375** |

*(1) Parity with Standard Hashing.* When the context is unedited, both key modules recover the correct keys step by step. Consequently, the $p$-value distributions align cleanly across mark modules, as demonstrated in Fig. 3. This parity is important to practical deployments; SIMKEY does *not* weaken detection or inflate false positives in benign settings.

*(2) Better Detectability Under Paraphrase Edits.* We find that SIMKEY is robust to meaning-preserving edits that alter surface forms while keeping semantics close to the original intent. Standard hashing, on the other hand, ties keys to exact recent tokens and thus loses detectability quickly. We can see this effect examining the *translated token replacement* attacks presented in Fig. 4. We observe consistent TPR@1%FPR gains and lower $p$-values across each of the three mark modules. SIMKEY's TPR gains appear mainly at higher token replacement levels where the attack is stronger.

*(3) Edits That Change Meaning Remove Watermark.* For unrelated replacements or insertions that shift topic or intent, watermarks using both SIMKEY and the baseline key degrade similarly, as evidenced by Table 1. This is the intended behavior of a semantically aware watermark; when semantics drift, the key should change, preventing harmful or off-topic additions from inheriting machine-generated attribution. Additionally, Table 1 shows that under related token replacements where the meaning of the text stays the same, SIMKEY performs *much* better.

*(4) Longer Generation and Conversational Perturbations.* When watermarking longer (500-token generations), for a fixed false positive rate (FPR) of 1%, SIMKEY improves true positive rate (TPR) across all three mark modules under both unrelated and related token substitutions (Table 2). This suggests that the benefits of semantic keying persist even when semantic drift accumulates over many sentences. Additionally, across 20 multi-turn conversations (see Appendix C), we obtain a TPR of 0.80 at 1% FPR, indicating that SIMKEY maintains reliable detection even when semantics evolve over several turns.

*(5) Does Not Add Distortion to the Sampling.* SIMKEY only replaces the key module. Because the mark module and base LM distribution are unchanged, the perplexity of the sampled text tracks the standard hashing almost exactly (Fig. 5). The primary driver of distortion remains the mark module itself: ExpMin is closest to unwatermarked text, SynthID induces some moderate change in perplexity, and WaterMax induces the greatest change. Using SIMKEY minimally affects the perplexity or this relative ordering, as expected.

*(6) Improved robustness to LLM-in-the-loop paraphrasing attack.* Before evaluating our LLM-in-the-loop paraphrasing attack, we quantify semantic drift over the paraphrases by computing BERTScore similarity between the original and paraphrased texts. Across 80 samples, the first paraphrase achieves a mean similarity of 0.71 ($\pm 0.10$), and the second paraphrase achieves 0.69 ($\pm 0.11$), indicating nontrivial shifts in meaning. Even under this aggressive attack, SIMKEY preserves more signal than standard hashing at a fixed 1% FPR: ExpMin with SIMKEY achieves a TPR of 0.13 compared to 0.063 with standard hashing, and WaterMax with SIMKEY achieves a TPR of 0.088 compared to 0.013. Although detectability decreases for all methods, these results suggest that semantic keying still improves resilience to strong paraphrasing.

Beyond the experiments presented in the main text, we include several additional evaluations. These cover (i) robustness across semantic embedders and languages (Appendix B.4), (ii) sensitivity to

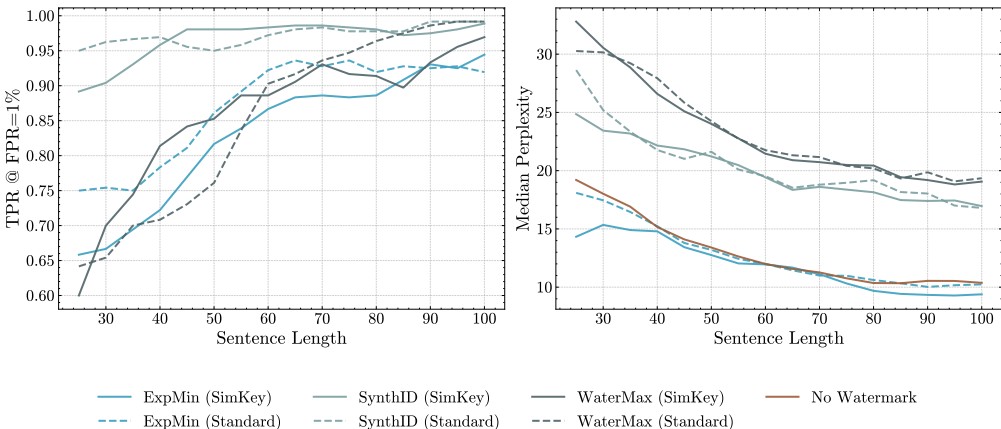

Figure 5: **SIMKEY preserves detectability for unmodified text, and the distribution of water-marked text**. (a) Detectability increases with sentence length for all mark modules, with SIMKEY and standard hashing performing similarly. (b) Perplexity depends on mark module: ExpMin is nearly indistinguishable from unwatermarked text, SynthID has higher perplexity, and WaterMax has the highest; with SIMKEY and standard hashing performing basically the same.

|  | Unrel-50tok | | Rel-50tok | |
|---|---|---|---|---|
| Method | St. Hash | SimKey | St. Hash | SimKey |
| ExpMin | 0.638 | **0.800** | 0.113 | **0.888** |
| SynthID | 0.800 | **0.813** | 0.888 | **0.925** |
| WaterMax | 0.400 | **0.563** | 0.138 | **0.788** |

Table 2: **Long-context robustness under token replacement.** TPR at 1% FPR for 500-token generations under 50 unrelated (Unrel-50tok) or related (Rel-50tok) token substitutions.

key-module hyperparameters (Appendix B.2), (iii) spoofing and forgery attempts (Appendix B.5), and (iv) an analysis of seed diversity in long generations (Appendix B.3). Together, these supplementary studies further support our central conclusion that SIMKEY reliably enhances robustness to meaning-preserving perturbations while preserving the underlying mark module.

## 5 LIMITATIONS

**Semantic Robustness for the Mark Module.** While our key module is responsible for determining the seed encodes the semantics of the text, the mark modules, which embeds the watermark into individual tokens, are not semantic in nature. That is, replacing a watermarked token with a synonymous alternative may significantly affect the likelihood of detecting the watermark for that token, even if the seed used for that token generation is correctly recovered. However, in sufficiently long text generations, it is likely that some words or tokens will remain unchanged after transformations such as translation to another language and back. In other words, SIMKEY makes the key schedule semantic, but the watermark signal at the token level is still governed by the particular mark module it is combined with. This is especially true for named entities such as people or places, or for punctuation marks, which will often be mapped back to their original token.

A natural extension to our work would be to introduce semantic awareness into the *mark module* as well. For example, one could design key-dependent preferences that favor semantically related words rather than specific surface forms. This would likely further improve the robustness of the watermarking scheme to synonym substitutions and similar removal attacks that we tested in this paper. However, such modifications would trade off with the distortion-free guarantees offered by existing approaches, which is a highly desirable property (see e.g., Section A, (Kuditipudi et al., 2024)). We leave an investigation of semantic mark modules to future work.

**Utilizing Additional Mark Modules**. We find SIMKEY to be compatible with most state-of-the-art watermarking techniques. Yet, certain mark modules may require additional adaptation. For example, "red-green" list approaches often adjust token probabilities by a fixed shift based on a hash of prior context (Kirchenbauer et al., 2024). When combined with the *key index variation* described

in Section 3.1, this can lead to unintended behavior: many tokens may appear on the green list for at least one index, *even* in unwatermarked text, thereby weakening detection. In such cases, SIMKEY can still be applied by disabling index variation, or with potentially other adaptations. More generally, we expect the method to extend naturally to a wide range of existing and future watermarking schemes.

**Privacy and deployment considerations.** In our design, semantic embeddings are computed locally from the already-visible context tokens and are never transmitted or logged as an additional side channel. Both during generation and detection, SIMKEY has access only to the same token sequence that any non-watermarked system would observe, and the embedding is used solely to derive an internal key. Thus, SIMKEY does not introduce a new communication or leakage path beyond standard autoregressive generation, and any privacy properties of the deployment are determined by the underlying LLM and logging policies rather than by the key module.

## 6 RELATED WORKS

**Watermarking Language Models.** Most LLM watermarking either perturbs the sampling distribution to embed detectable signals or aims to preserve it while enabling detection. Kirchenbauer et al. (2024) partition tokens at each step into pseudo-random "green"/"red" sets via a hash of prior tokens. Unigram (Zhao et al., 2023b) fixes these lists for robustness, but such approaches face spoofing risks (Liu et al., 2025; Jovanović et al., 2024; Sadasivan et al., 2023). Related methods include Gumbel-Soft (Fu et al., 2024), Duwak (Zhu et al., 2024), SWEET (Lee et al., 2024), and NS-Watermark (Takezawa et al., 2025). SynthID-text (Dathathri et al., 2024) similarly adjusts sampling to preserve quality and latency. See Liu et al. (2025) for a comprehensive review.

**Distortion-Free LLM Watermarking.** Aaronson (2023) augment the exponential mechanism with a hash of prior tokens. Christ et al. (2023) use cryptographic indistinguishability, making detection without a key computationally hard. Exponential Minimum Sampling (ExpMin) (Kuditipudi et al., 2024) seeds Gumbel-Softmax with a pseudo-random sequence, leaving the per-token distribution unchanged; detection then correlates text with the sequence, but removal attacks remain a challenge. Our module replaces fixed seeds (e.g., token-hash or PRNG) with a dynamic key derived via semantic hashing of context, thus addressing a major challenge framed by the authors of ExpMin.

**Semantic Watermarking.** To resist surface edits, semantic methods embed signals tied to meaning. Liu et al. (2024a) use an external encoder (e.g., BERT) and a learned watermark head, but require training. Other semantic approaches include Remark-LLM (Zhang et al., 2024b), SemStamp (Hou et al., 2024a), and $k$-SemStamp (Hou et al., 2024b). In contrast, SIMKEY is a *key module* rather than a full semantic watermarking pipeline: it does not impose additional semantic constraints on the token distribution itself, but instead ensures that the internal key used by an existing mark module varies smoothly with the meaning of the context. This makes SIMKEY complementary to prior semantic watermarking schemes. In principle, a semantic mark module could also be paired with SIMKEY to obtain both a semantic key schedule and semantic token-level constraints, though we leave such combinations to future work. Our approach follows this direction without model-level changes: it derives a local semantic key from context to enable semantic awareness, preserve analytical tractability, and remain compatible with state-of-the-art mark modules (Huang & Wan, 2024); this is desirable in that SIMKEY can be readily integrated into existing state of the art watermarking schemes without heavy adaptations.

## 7 CONCLUSION

We present SIMKEY, a key module that utilizes Locally Sensitive Hashing to allow embedding of detectable and robust watermarks in language model outputs. Our key identity remains stable under edits that preserve semantics, but not under edits that change them. Through experimental evaluations, we show that our module improves robustness against various attacks while maintaining generation perplexity comparable to the same watermarking methods using existing key modules. These results highlight the potential of semantic-aware keys for watermarking as a practical and principled solution for practical and responsible deployment of language models.

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

## A  EXPONENTIAL MINIMUM SAMPLING

Exponential Minimum Sampling enables randomized token selection based on the LLM's probabilities in a numerically stable way, while also exhibiting properties that make it effective for watermarking. Let $\mathbf{p} \in [0,1]^V$ be a distribution over the vocabulary. Suppose $\xi \in [0,1]^V$ is a random variable where each entry is independently drawn from the uniform distribution on $[0,1]$. Exponential minimum sampling selects the next token via

$$i^* \leftarrow \underset{i \in \{1,\ldots,|V|\}}{\arg\min} \frac{-\log([\xi]_i)}{p_i}. \tag{5}$$

**Lemma A.1** (Exponential Minimum Sampling). *The probability that a token $i^*$ is selected via exponential minimum sampling in Equation 5 is:*

$$\Pr(i^* \text{ is selected}) = p_{i^*}$$

This is a well-known fact that follows from Gumbel sampling see e.g., Kuditipudi et al. (2024). For the interested reader, we present a proof below.

*Proof.* The first observation is that each term in the minimum is an exponentially distributed random variable with rate $p_i$. To see this, notice that $-\log(\cdot)$ applied to a uniform variable results in an exponentially distributed random variable with rate 1, i.e., $-\log([\xi]_i) \sim \text{Exp}(1)$. Next, observe that dividing an exponentially distributed variable by a constant multiplies its rate by the constant i.e., $\frac{-\log([\xi]_i)}{p_i} \sim \text{Exp}(p_i)$. We can then directly analyze the probability that a particular $i^*$ achieves the minimum value. For notational convenience, let $X_i = \frac{-\log([\xi]_i)}{p_i}$. Then $X_i \sim \text{Exp}(p_i)$ and

$$\Pr\left(i^* = \underset{i \in \{1,\ldots,|V|\}}{\arg\min} \frac{-\log([\xi]_i)}{p_i}\right) = \int_{x=0}^{\infty} \Pr(X_{i^*} = x) \Pr(\forall_{i \neq i^*} X_i > x) dx$$

$$= \int_{x=0}^{\infty} p_{i^*} e^{-p_{i^*} x} \left(\prod_{i \neq i^*} e^{-p_i x}\right) dx = p_{i^*} \int_{x=0}^{\infty} e^{-(p_1 + \ldots p_V)x} dx = p_{i^*}$$

where the second equality follows by plugging in the PDF and CDF of the exponential distribution. The statement immediately follows. $\qquad\square$

### A.1  CLOSED-FORM $p$-VALUE UNDER EXPMIN

Recall from Section 3.2 that at position $t$ we evaluate all $k$ candidate key indices and take the per-token alignment cost as the minimum across candidates. Under ExpMin, for a fixed token $y_t$ and key index $j$, the quantity,

$$Z_{t,j} := -\log([\xi_{t,j}]_{y_t}) ,$$

is exponentially distributed with rate 1. This is because $[\xi_{t,j}]_{y_t} \sim \text{Unif}[0,1]$. Taking the min across $k$ independent candidates then yields,

$$C_t := \min_{j \in \{1,\ldots k\}} Z_{t,j} \sim \text{Exp}(k) ,$$

with $Pr[C_t > c] = e^{-kc}$.

If we assume independence across positions[3], then the total alignment cost over the $n$ tokens is,

$$S_n := \sum_{t=1}^{n} C_t ,$$

---

[3]This holds exactly if the seeds at different positions are independent; in practice it is an accurate approximation because the per-position seeds are (pseudo)random functions of the preceding context and key index.

has a Gamma distribution with the shape $n$ and rate $k$ i.e. $S_n \sim \mathrm{Gamma}(\mathrm{shape} = n, \mathrm{rate} = k)$ (Ross, 2023). Its CDF admits the closed form,

$$F_{S_n}(s) = Pr[S_n \leq s] = \frac{\gamma(n, ks)}{\Gamma(n)} = 1 - e^{-ks} \sum_{m=0}^{n-1} \frac{(ks)^m}{m!} \, .$$

Because lower costs are stronger evidence of watermarking, the one-sided $p$-value is,

$$p = F_{S_n}(s_{obs}) = 1 - e^{-ks_{obs}} \sum_{m=0}^{n-1} \frac{(ks_{obs})^m}{m!} \, .$$

However, in our implementation, we report the *mean* cost $\bar{S}_n := S_n/n$ instead of the sum. Since, $\bar{S}_n \sim \mathrm{Gamma}(\mathrm{shape} = n, \mathrm{rate} = kn)$, the corresponding CDF is,

$$F_{\bar{S}_n}(a) = Pr[\bar{S}_n \leq a] = \frac{\gamma(n, kna)}{\Gamma(n)} = 1 - e^{-kna} \sum_{m=0}^{n-1} \frac{(kna)^m}{m!} \, ,$$

which means the $p$-value is $p = F_{\bar{S}_n}(a_{obs})$. This is equivalent to the *sum* up to the deterministic scaling by $n$.

## B  ADDITIONAL RESULTS

### B.1  DETECTABILITY.

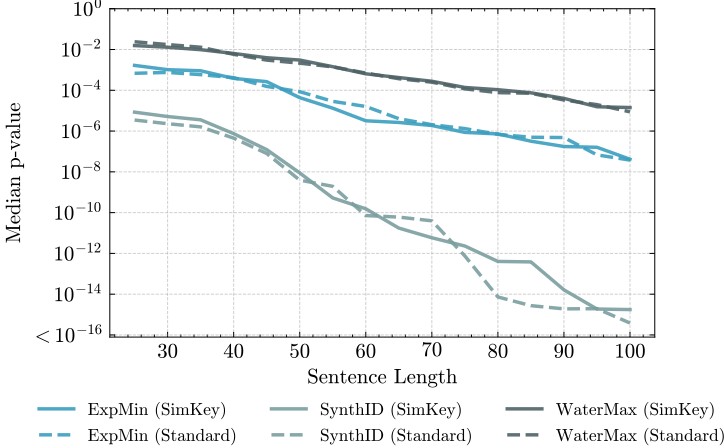

Figure 6: The median $p$-value among 80 generated texts for each sentence length. SIMKEY preserves the detectability, with the median $p$-value staying basically the same between SIMKEY and standard hashing.

## B.2 HYPERPARAMETER ABLATION OVER $b$ AND $k$

Figure 7 visualizes the sensitivity of SIMKEY to the number of SimHash bits $b$ and key indices $k$ as heatmaps for ExpMin and WaterMax. We study this by sweeping over a grid of configurations and measuring TPR at 1% FPR. We observe that very small configurations such as $(b, k) = (2, 2)$ can produce strong detection signatures but noticeably degrade fluency, likely due to frequent key collisions that make sampling more deterministic. In contrast, the configuration used in our main experiments, $(b, k) = (4, 4)$, lies on a natural Pareto frontier: it provides substantial robustness gains while keeping perplexity and distributional distortion nearly identical to standard hashing. Increasing either $b$ or $k$ much further tends to reduce robustness, because keys become overly sparse and less likely to match under semantic perturbations.

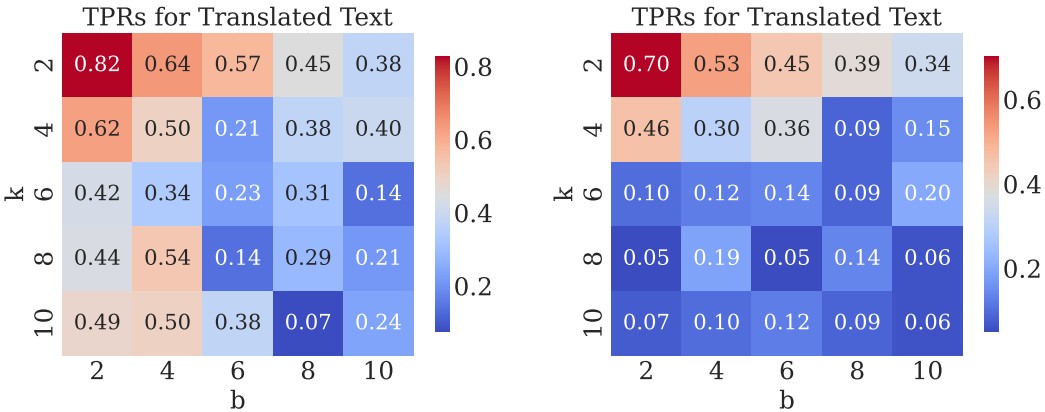

Figure 7: **Sensitivity of SIMKEY to SimHash bit-width $b$ and key-index count $k$.** TPR at 1% FPR across a grid of $(b, k)$ configurations for **ExpMin** (left) and **WaterMax** (right). The $(b, k) = (4, 4)$ configuration used in the main text lies near a Pareto frontier between robustness and fluency.

## B.3 KEY DIVERSITY IN LONG GENERATIONS

To better understand how often SIMKEY reuses the same key in long-context settings, we examine the diversity of seeds produced during 500-token generations. For a representative sample, we record all seeds used across a single 500-token continuation and count how many times each distinct seed occurs. Table 3 summarizes the resulting multiplicity distribution. Although some seeds do repeat, as intended, since nearby semantics should often produce the same key, more than half of the seeds appear only once, and nearly 80% appear at most twice. This indicates that the key space remains diverse in practice, while still allowing nearby contexts to share keys and thus preserve detectability under meaning-preserving edits.

Table 3: Multiplicity of distinct seeds produced by SIMKEY in a representative 500-token generation.

| Occurrences per seed | Number of seeds | Percentage of seeds |
|---|---|---|
| 1 | 128 | 52.03% |
| 2 | 60 | 24.39% |
| 3 | 29 | 11.79% |
| 4 | 13 | 5.28% |
| 5 | 6 | 2.44% |
| $\geq 6$ | 9 | 3.66% |

## B.4 ROBUSTNESS ACROSS EMBEDDERS AND LANGUAGES

Our main experiments use all-`MiniLM-L6-v2` as the semantic encoder, but SIMKEY is not tied to a specific embedding model. To test this, we replace the encoder with `BAAI/bge-small-en` and

Table 4: **Robustness under an alternative semantic encoder.** TPR at 1% FPR under 10 unrelated (Unrel-10tok) or related (Rel-10tok) token substitutions when replacing `all-MiniLM-L6-v2` with `BAAI/bge-small-en`. SIMKEY consistently improves robustness relative to standard hashing.

| Method | Unrel-10tok | | Rel-10tok | |
| --- | --- | --- | --- | --- |
| | St. Hash | SimKey | St. Hash | SimKey |
| ExpMin | 0.500 | 0.863 | 0.250 | 0.950 |
| SynthID | 0.950 | 0.975 | 0.825 | 0.988 |
| WaterMax | 0.450 | 0.813 | 0.113 | 0.800 |

Table 5: **Multilingual robustness on French text.** TPR at 1% FPR under 10 unrelated (Unrel-10tok) or related (Rel-10tok) token substitutions for French generations. SIMKEY typically improves robustness relative to standard hashing.

| Method | Unrel-10tok | | Rel-10tok | |
| --- | --- | --- | --- | --- |
| | St. Hash | SimKey | St. Hash | SimKey |
| ExpMin | 0.475 | 0.788 | 0.250 | 0.713 |
| SynthID | 0.975 | 0.988 | 0.713 | 0.963 |
| WaterMax | 0.463 | 0.313 | 0.138 | 0.338 |

repeat the token-replacement robustness experiments, again reporting TPR at 1% FPR. As shown in Table 4, SIMKEY retains the same qualitative behavior: it consistently improves robustness under both unrelated and related token substitutions across ExpMin, SynthID, and WaterMax.

We also examine a multilingual setting by generating and perturbing French text using the same pipeline (generation, unrelated and related token substitutions). Table 5 shows that SIMKEY generally improves robustness in French as well, suggesting that the benefits of semantic keying are not confined to English-only settings or a particular encoder.

## B.5 SPOOFING AND FORGERY ATTEMPTS

Finally, we examine a spoofing scenario in which an adversary attempts to synthesize text that appears watermarked without access to the secret salt. We provide a strong external LLM with a set of genuine SIMKEY-watermarked examples and a matched set of unwatermarked examples, and instruct it to produce new samples that "match whatever hidden pattern" might distinguish the two groups, while avoiding direct copying or paraphrasing. We then mix these forged samples with genuine watermarked outputs and run our detector blindly. On a test set of 80 texts (40 genuine, 40 forgeries), calibrated at a 1% FPR on human text, our detector correctly identifies all 40 genuine watermarked samples and rejects all 40 forgeries. This indicates that mimicking stylistic properties alone is insufficient to reproduce the internal key schedule induced by the secret salt and semantic SimHash.

## C  COMPLETE CONVERSATIONAL QUESTION PROMPTS

| | |
|---|---|
| **Topic 1: Making sourdough bread** | 1. How should I feed and maintain a sourdough starter I keep in the fridge? 

 2. If my starter feels sluggish, how can I adjust hydration to get a crustier loaf? 
 3. If I'm tired of sourdough, how can I switch to a simple yeasted rustic loaf? |
| **Topic 2: Choosing a running shoe** | 1. What kind of running shoes are best if I overpronate a little on long runs? 

 2. When can I safely transition from stability shoes to lighter, more flexible trainers? 
 3. Could you outline a simple first-time 5K training plan for a casual runner? |
| **Topic 3: Caring for houseplants** | 1. How often should I water a pothos in a bright room so I don't overwater it? 

 2. What should I do if my pothos leaves start yellowing, especially near the base? 
 3. How can I design a small indoor plant shelf that keeps low-light plants happy? |
| **Topic 4: Explaining lunar eclipses** | 1. Why does the Moon sometimes turn red during a lunar eclipse? 

 2. How can I tell in advance if a lunar eclipse will be visible from my city? 
 3. Any tips for photographing night-sky events like eclipses with a phone? |
| **Topic 5: Brewing pour-over coffee** | 1. What grind size should I use for a balanced pour-over coffee? 

 2. How should I adjust bloom time if the coffee tastes sour or under-extracted? 
 3. How do I choose between light and medium roasts for brighter flavors? |
| **Topic 6: Basic bicycle maintenance** | 1. How can I fix a squeaky bike chain without taking the bike to a shop? 

 2. What's the simplest way to align rim brakes that rub on one side? 
 3. How would you plan a short, beginner-friendly weekend cycling route? |
| **Topic 7: Writing a short mystery story** | 1. How can I quickly sketch a memorable detective character? 

 2. What's an easy way to add a red herring without confusing readers? 
 3. How do I structure a twist ending that feels surprising but still fair? |
| **Topic 8: Packing for a day hike** | 1. What are the essentials for a three-hour day hike? 

 2. How do I choose a lightweight water filter for short hikes? 
 3. How can I find less crowded alternatives to popular trails? |
| **Topic 9: Building a simple budget** | 1. How should I set up a basic monthly budget? 

 2. What's a good way to monitor and reduce grocery expenses? 
 3. How can I save a small monthly amount for a weekend trip? |
| **Topic 10: Taking care of a senior dog** | 1. How can I help my senior dog with stiff joints move comfortably? 

 2. Should I adjust feeding schedules as my dog becomes less active? 
 3. What features matter most when choosing a bed for an older dog? |

| | |
|---|---|
| **Topic 11: Learning basic guitar chords** | 1. Easiest way to switch cleanly between G and C chords? |
| | 2. A simple strumming pattern to practice with those chords? |
| | 3. Ultra-easy songs that mainly use G and C? |
| **Topic 12: Improving sleep habits** | 1. How can I set a consistent bedtime with a chaotic schedule? |
| | 2. How to reduce screen time before bed without going cold turkey? |
| | 3. How to design a short, relaxing morning routine? |
| **Topic 13: Making homemade salad dressing** | 1. What's a reliable oil-to-vinegar ratio? |
| | 2. How can I add herbs/spices for a Mediterranean flavor? |
| | 3. Which dressings pair best with delicate vs. sturdy greens? |
| **Topic 14: Starting a simple workout routine** | 1. Which beginner bodyweight exercises should I start with? |
| | 2. When and how should I add light dumbbells? |
| | 3. How can I track progress without getting obsessed with numbers? |
| **Topic 15: Cooking scrambled eggs** | 1. Difference between whisking and folding when scrambling eggs? |
| | 2. Why cook scrambled eggs over low heat, and how low is "low"? |
| | 3. What easy variations make scrambled eggs more interesting? |
| **Topic 16: Understanding electric cars** | 1. How does regenerative braking recover energy? |
| | 2. Pros and cons of home vs. public charging? |
| | 3. How should I plan road trips around charging stations? |
| **Topic 17: Taking photos with a smartphone** | 1. How can I use natural light to improve photos? |
| | 2. What is the rule of thirds and how do I apply it quickly? |
| | 3. Simple editing tweaks that improve phone photos? |
| **Topic 18: Playing chess openings** | 1. Basic ideas behind the Italian Game for White? |
| | 2. Typical tactics to watch for in the first few moves? |
| | 3. How do I transition into a sensible middlegame plan? |
| **Topic 19: Organizing a small birthday dinner** | 1. How can I pick a simple, stress-free menu? |
| | 2. Inexpensive decoration ideas that still feel festive? |
| | 3. How should I write short, friendly invitations? |
| **Topic 20: Caring for cast-iron cookware** | 1. How do I season a cast-iron skillet from scratch? |
| | 2. Best way to clean it without stripping seasoning? |
| | 3. Which meals help improve seasoning as I cook? |

# D PROMPTS USED FOR GENERATION AND EXPERIMENTAL DETAILS

**Experimental Details** Our experiments do not rely on external datasets but instead use a fixed collection of short prefix prompts (listed in Table 6) to initiate generation in a controlled manner. Incorporating SIMKEY adds only modest computational overhead: for 500-token generations, runtime increases by roughly 10-15% (e.g., ExpMin: 1.95s → 2.22s; WaterMax: 2.03s → 2.28s), while detection exhibits a somewhat larger slowdown due to recomputing semantic embeddings over the entire output (ExpMin: 0.60s → 1.57s; WaterMax: 0.62s → 1.56s). These costs stem from SimHash and embedding computations rather than any change to model forward passes, and remain small relative to typical inference latency. All generations and detections were performed with a Llama-3.2-3B model on an AWS g6e.2xlarge instance equipped with an NVIDIA L40S GPU.

| General-Purpose Generation Prompts | |
|---|---|
| Once upon a | The sun rises |
| She suddenly realized | He never imagined |
| In the distance | Without a doubt |
| The clock ticked | A strange sound |
| Under the moonlight | Beyond the horizon |
| It all began | Nobody expected that |
| Suddenly, she gasped | He walked away |
| They finally arrived | The air smelled |
| Through the mist | The storm raged |
| A loud crash | She carefully stepped |
| On the table | The night fell |
| It was dark | With a sigh |
| A gentle breeze | In the morning |
| A voice whispered | The silence broke |
| Over the hills | She quickly turned |
| He nervously glanced | The shadows moved |
| With trembling hands | A soft melody |
| The door creaked | In an instant |
| She held on | He stood still |
| The wind howled | In the garden |

Table 6: Short prefix prompts used for general-purpose generation experiments.

# E USE OF LLMS

We used LLMs to polish the writing and find typos.

