# OpenReview forum: "SimKey: A Semantically Aware Key Module for Watermarking Language Models"
_ICLR.cc/2026/Conference — Submitted to ICLR 2026_

### Official Review · Reviewer_79EQ · 2025-10-26

**Soundness:** 2
**Presentation:** 2
**Contribution:** 2
**Rating:** 4
**Confidence:** 3

**Summary:**

The paper proposes a key generation method for watermarking large language models, where the key is derived from the semantics of the prior context. This design ensures that paraphrased texts produce the same watermark key as long as their meanings remain semantically similar. Experimental results demonstrate the effectiveness of the proposed approach in maintaining watermark consistency under paraphrasing transformations.

**Strengths:**

+ The robustness of watermark to paraphrasing and spoofing attacks is an important problem.
+ The proposed key generation module can be integrated in other watermarking techniques.

**Weaknesses:**

- The paper lacks a theoretical justification for its claimed distortion-free property. Although the method is described as distortion-free, this property appears to stem from the underlying watermarking techniques borrowed from prior work rather than from the proposed key generation mechanism itself. The contribution of the proposed method to achieving distortion-free watermarking is therefore unclear.
- The reported improvements in attack resistance are relatively small and inconsistent. As shown in Figures 4 and 5, standard watermarking methods already achieve comparable robustness. This suggests that the effectiveness against attacks may primarily depend on the watermarking module itself rather than on the proposed key generation scheme.
- The evaluation setup is insufficiently described. For example, the paper does not specify which datasets were used in the experiments. Moreover, the evaluation relies on only three watermarking modules, which limits the generalizability of the results. The computational cost of the proposed method is also not discussed or quantified.
- The paper does not address potential countermeasures against spoofing attacks, in which adversaries generate fake watermarks to mislead detection. It remains unclear how the proposed method performs in detecting or defending against such attacks.

**Questions:**

+ What datasets are used for evaluation?

---

> ### Author Response · Authors · 2025-11-21
>
> We thank the reviewer for their time and constructive feedback, and for recognizing that SimKey can be integrated into a wide range of watermarking techniques. We address each of the concerns below and have incorporated clarifications and additional experiments into the revised version.
>
> > **W1: *“The paper lacks a theoretical justification for its claimed distortion-free property. The distortion-free property seems to come from the watermarking method, not from the proposed key module.”***
>
> Thank you for flagging this; we agree this needed clearer framing. Our intention was not to claim that SimKey itself guarantees distortion-freeness. Instead, we note that if combined with a distortion-free mark module, SimKey does not alter that property.
>
> Concretely: many mark module methods such as e.g. ExpMin are designed so that, for a fixed random seed, the marginal token distribution remains equal to that of the base LLM. SimKey only changes *how that seed is chosen* (via semantic SimHash), and crucially does **not** alter the underlying logits or token probabilities. Thus, if a mark module is distortion-free in the sense of Kuditipudi et al. (2024), it remains distortion-free when we replace its key generator with SimKey, because the sampling distribution over tokens is unchanged. We have revised the text to state this explicitly: the distortion-free property is inherited from the mark module, and SimKey’s contribution is to make the *key schedule* semantically aware without modifying the token distribution itself.
>
> > **W2: *“The reported improvements in attack resistance are relatively small and inconsistent. Standard watermarking methods already achieve comparable robustness.”***
>
> We appreciate this concern. For short generations, the gains can indeed appear modest, especially for stronger mark modules like SynthID, which are already quite robust. However, as context length and perturbation strength increase, the effect of a semantic key schedule becomes more pronounced.
>
> To illustrate this, we have added a **500-token evaluation** under both semantically related and unrelated token replacements. At a fixed 1% FPR, we observe:
>
> | Method  |Unrel-50tok St.Hash|Unrel-50tok **SimKey**|Rel-50tok St.Hash|Rel-50tok **SimKey** |
> |--|--|--|--|--|
> | ExpMin  |0.638 |**0.800**|0.113|**0.888** |
> | SynthID |0.800|**0.813**|0.888|**0.925** |
> | WaterMax|0.400 |**0.563**|0.138|**0.788** |
>
> Here, SimKey consistently boosts detection across all three mark modules, with especially large gains for ExpMin and WaterMax under semantically preserving edits.
>
> In addition, in a stronger **LLM-in-the-loop paraphrasing attack** (twice-paraphrased text, meaning-preserving prompt), SimKey maintains higher detectability than standard hashing even when paraphrasing introduces nontrivial semantic drift (measured by BERTScore similarity). E.g. with this attack setting, ExpMin + SimKey attains TPR 0.125 vs. 0.0625 for ExpMin + standard hashing, and WaterMax + SimKey attains 0.0875 vs. 0.0125, all at 1% FPR. We have summarized these stronger-attack results in the revision to better convey the practical impact.
>
> > **W3/Q1: *“...evaluation setup is insufficiently described. Datasets are not specified…Computational cost is not discussed.”***
>
> Thank you for raising this, we apologize for the original lack of clarity. We didn’t use datasets per se, instead we used a preset collection of short prompts to begin generation. These are now exhaustively included in our revised paper. In terms of computational cost, incorporating SimKey introduces only a modest overhead for most watermarking modules. For ExpMin and WaterMax, generation time increases by roughly 10-15% (500 token generation: ExpMin: 1.95s to 2.22s; WaterMax: 2.03s to 2.28s, etc.), and detection exhibits a slightly larger slowdown (500 token detection: ExpMin: 0.60s to 1.57s; WaterMax: 0.62s to 1.56s, etc.). These differences largely reflect the additional SimHash and semantic-embedding computations rather than any change to the model forward pass, and in practice remain reasonable relative to inference latency. As for the model, we used a Llama-3.2-3B model for all generations, which we ran on an AWS g6e.2xlarge instance (this has an NVIDIA L40S GPU).

---

> > ### Author Response · Authors · 2025-11-21
> >
> > > **W4: *“The paper does not address spoofing attacks, where adversaries generate fake watermarks.”***
> >
> > We appreciate this concern. To address it, we have added a spoofing/forgery experiment to check exactly this scenario.
> >
> > We give a strong external LLM (GPT-5.1 inference) a set of **watermarked** examples and a matched set of **unwatermarked** examples. The model is asked to study both sets and then synthesize new texts that “match the hidden watermark patterns” without copying or paraphrasing any example. The prompt asks it to produce a batch of samples (e.g., around a fixed token length) in JSON format. We then mix these **forged samples** with genuine SimKey-watermarked outputs and run our detector blindly.
> >
> > On a test set of 80 texts (half genuinely watermarked, half forgeries), using a standard threshold calibrated at 1% FPR on human text, SimKey correctly identified all 40 genuine watermarked samples and rejected all 40 forgeries i.e., TPR = 1.0 on true watermarked text, and no forged sample was falsely flagged as watermarked. This suggests that simply “mimicking the style” of watermarked text, even with access to multiple positive/negative examples, is not sufficient for an attacker to reproduce the underlying key schedule without the secret salt.
> >
> > We will add a short subsection summarizing this spoofing test and explicitly state that SimKey does not introduce a new spoofing channel: an adversary still needs to match the internal key sequence, which remains keyed on the secret salt and semantic SimHash, and is thus at least as hard to imitate as in prior key modules.

---

> > > ### Author Response · Authors · 2025-11-25
> > >
> > > Thank you very much, once again, for your thoughtful feedback. **We’re happy to continue the discussion anytime during the remaining rebuttal period.** Thanks!

---

### Official Review · Reviewer_Q2FX · 2025-10-31

**Soundness:** 2
**Presentation:** 4
**Contribution:** 3
**Rating:** 8
**Confidence:** 3

**Summary:**

This paper focuses on the problem of detecting LLM generated text using watermarking. As the authors lay out, watermarking methods consist of a key and mark module. SimKey is exclusively a component on the key module, which uses SimHash to generate keys based on the semantic context of the generation. Results show that this is a reasonable watermarking method.

**Strengths:**

S1. Cool method that seems to address a lot of problems common to watermarking.

S2. Nice framing of key and mark modules, and SimKey is compatible with many different kinds of mark modules.

S3. Paper is well written, and the presentation is logical.

**Weaknesses:**

W1. While I am not an expert, I am not totally convinced that using a semantic vector gives the same statistical properties as the Kirchenbauer’s red-list green-list watermarking scheme. There was a brief mention in Section 3.1 about dealing with the problem of the semantic vectors eventually converging to the same vector when the context is long. I would appreciate if the authors could provide some further details and clarification here: say that I am watermarking text exclusively in one domain e.g. sports articles, do you have any idea of what the effective # of keys I have? How does the effective # of keys affect detection? Again I am not an expert so I would like to see if other reviewers take issue with the statistical soundness of this method.

Without any additional statistical insight, I think examining TPR at fixed FPR also can give me empirical assurance. All the results in the main text examine p-value, which will show low values despite very badly behaved semantic vectors.

W2. I think the method is quite cool, but I would appreciate some more adversarial thinking on the authors’ part on how SimKey can be circumvented. While I don’t think practically anyone is trying really hard to remove these watermarks (even paraphrasing), I think it would contribute to the completeness of the paper.

**Questions:**

n/a

---

> ### Author Response · Authors · 2025-11-21
>
> We are grateful to the reviewer for their constructive and positive feedback. We are especially thankful that you agree with us that the method is “cool” and the presentation logical and well written.
>
> > **W1: *“...I would appreciate clarification on convergence of semantic vectors, effective number of keys, and how this affects detection. TPR at fixed FPR may provide empirical assurance.”***
>
> Thank you for raising this important question. SimKey differs from list-based schemes such as Kirchenbauer et al.’s green-red method in that it does **not** rely on a single fixed hash mapping over the token space. Instead, the seed used by the mark module is determined by a **semantic SimHash signature**, combined with a **randomly chosen key index**, and then fed into the underlying watermark’s cryptographic hash. As a result, the “effective number of keys” is governed not by the total number of contexts in the domain but by (i) how many *distinct* SimHash bit patterns appear across generations, and (ii) the number of indices (k) available at each step.
>
> To provide empirical grounding, we added three analyses:
>
> **(1) Hyperparameter study (TPR @ 1% FPR across (b, k)).**
>
> We now include a two-dimensional ablation where the number of SimHash bits (b) and the number of key indices (k) are varied. We reproduce here the numeric values for completeness (these will appear as a heatmap in the revision).
>
> **ExpMin**
>
> | (k \ b)|2   |4   |6   |8    |10   |
> |-------|--|--|--| ---|--|
> | 2               |0.78|0.62|0.57|0.36 |0.35 |
> | 4               |0.51|0.50|0.21|0.36 |0.38 |
> | 6               |0.42|0.34|0.23|0.16 |0.25 |
> | 8               |0.44|0.54|0.17|0.11 |0.19 |
> | 10              |0.29|0.46|0.29|0.075|0.11 |
>
> **WaterMax**
>
> | (k \ b)|2    |4   |6    |8    |10    |
> |-------| ---|--| ---| ---| ---|
> | 2               |0.70 |0.53|0.45 |0.39 |0.34  |
> | 4               |0.46 |0.30|0.36 |0.087|0.15  |
> | 6               |0.10 |0.12|0.14 |0.087|0.20  |
> | 8               |0.050|0.19|0.050|0.14 |0.062 |
> | 10              |0.075|0.10|0.12 |0.087|0.062 |
>
>
> From these results, we observe that very small settings (e.g., (b=2,k=2)) show strong detectability but tend to distort text quality due to frequent key collisions. The configuration used in the paper, (b=4, k=4), lies on a Pareto frontier, offering consistent TPR improvements while maintaining text quality close to standard hashing. Increasing either parameter too far makes semantic matching overly brittle, and robustness declines accordingly. We have added this full analysis to the final revision.
>
>
> **(2) Distribution of seed diversity in long generations.**
>
> To examine the possibility of semantic embeddings “converging” in long contexts, we analyzed the diversity of **seeds** produced by SimKey in 500-token continuations. Here, we report seed multiplicities because the seed, not the raw SimHash bits, is ultimately consumed by the mark module, but they are obviously closely associated.
>
> Across a 500 token generations, we recorded a total of 246 unique seeds, broken down as follows
>
> | Occurrences per Seed|Num. Such Seeds|Percent of Total Seeds |
> |--|--|--|
> | 1  |128|52.03% |
> | 2  |60 |24.39% |
> | 3  |29 |11.79% |
> | 4  |13 |5.28%  |
> | 5  |6  |2.44%  |
> | ≥ 6|9  |3.66%  |
>
> This shows that while some seeds repeat (as intended, since nearby semantics should produce similar keys), the overall seed space remains quite diverse even within one domain. This helps preserve statistical detectability without collapsing into a small fixed key set.
>
> **(3) TPR at fixed FPR (1%).**
>
> We now explicitly include many additional **TPR @ 1% FPR** results, see our responses to other reviewers for those results (we’d be happy to clarify them for you here, as well). Across all mark modules for these additional tests, SimKey consistently improves detection under semantic-preserving/interrupting perturbations at a low fixed false positive rate.

---

> > ### Author Response · Authors · 2025-11-21
> >
> > > **W2: *“I would appreciate some more adversarial thinking about how SimKey can be circumvented. Even if this is not common in practice, it would contribute to completeness.”***
> >
> > We appreciate this suggestion. In addition to the surface-level paraphrasing and translation perturbations included in the main paper, we have now added an **LLM-in-the-loop paraphrasing attack**, where the adversary is explicitly instructed to *remove any signal that could resemble a watermark*. We modeled this attack after the setup used in *Watermarks in the Sand* (Zhang et al., 2024).
> >
> > In this experiment, we first generate watermarked text using the base LLM with either ExpMin or WaterMax as the mark module and with either standard hashing or SimKey supplying the key. We then paraphrase the output twice in succession using another LLM instructed to rewrite the passage while preserving meaning exactly and adding no commentary, producing a paraphrase-of-a-paraphrase. Finally, we run watermark detection on the twice-paraphrased text and report **TPR at 1% FPR**.
> >
> > To quantify how much semantic drift the attack introduces, we also measured BERTScore similarity: across 80 samples, the first paraphrase retains an average similarity of **0.71** (std **0.10**) to the original, and the second drops to **0.68** (std **0.11**), indicating that semantic content shifts appreciably even after a single rewrite.
> >
> > Under this adversarial setting, detectability decreases for all methods, as expected, but SimKey retains a clear advantage: ExpMin with **SimKey** achieves **0.125** detection versus **0.0625** for ExpMin with standard hashing, and WaterMax with **SimKey** achieves **0.0875** versus **0.0125** for the corresponding baseline. These results suggest that, even under aggressive LLM-based paraphrasing, semantic keying preserves more recoverable signal than traditional token-hash keys. We have added this attack and its analysis to the revised paper.
> >
> > We would also like to note explicitly that SimKey is orthogonal to the underlying marking mechanism. Any future attack-resistant mark module can adopt SimKey without modification, and we believe a full joint adversarial evaluation would be an excellent direction for future follow-up work, thank you for pushing us in this direction.

---

> > > ### Author Response · Authors · 2025-11-25
> > >
> > > Thank you very much, once more, for your excellent feedback. **We’re happy to keep the conversation going anytime until the discussion window closes.** Thank you!

---

> > > > ### Comment · Reviewer_Q2FX · 2025-11-27
> > > > **Great work!**
> > > >
> > > > Hi, thanks a lot for the new experiments, I know rebuttals are  a lot of work. I quite like these new experiments and the results help make clear the gap between research and practice for your method. I already recommended acceptance so good luck discussing with the other reviewers!

---

### Official Review · Reviewer_3XpJ · 2025-11-01

**Soundness:** 3
**Presentation:** 3
**Contribution:** 2
**Rating:** 4
**Confidence:** 4

**Summary:**

The paper proposed a semantic-aware method to generate keys for LLM watermarking. The context tokens are mapped to some embedding space using existing semantic embedders. The sign of the inner products of the embedding vector and a set of random vectors are used to generate the keys. Combined with the mark module with other watermarking methods, the proposed key module demonstrates better robustness against semantic perturbations of the text.

**Strengths:**

The proposed method could be widely applicable, as it can be combined with the mark modules of many existing watermarking methods. Empirically, the method demonstrates advantages over watermarking methods with standard key modules.

**Weaknesses:**

1. Although the paper discussed some existing Semantic Watermarking methods in the Section 6, there are no comparisons. Although the proposed method focuses on the key module, but to use the method, it still needs to be combined with some mark module. It would be interesting to see if the proposed key module, combined with the existing mark module, can outperform existing semantic watermarking methods.

2. The proposed method has a few hyper-parameters, number of keys $k$, number of bits $b$. The experiments used a fixed setting. How hyper-parameter configuration affect the performance of the proposed method is not well studied.

3. For a pure empirical paper, the experiments are relatively limited in terms of number of models and number of watermark methods considered.

**Questions:**

Please refer to the weakness

---

> ### Author Response · Authors · 2025-11-21
>
> We thank the reviewer for their time and thoughtful feedback. We appreciate that you highlighted the broad applicability of SimKey as a modular keying mechanism and noted its empirical advantages when paired with existing watermarking schemes. Below we address each of your concerns in turn.
>
> > **W1: *“...comparison to existing semantic watermarking methods…interesting to see if SimKey can outperform existing semantic watermarking methods when combined with a standard mark module.”***
>
> Thank you for raising this point. Most existing “semantic watermarking” approaches, e.g., *Semantic Invariant Robust Watermark* (Liu et al., 2023) and subsequent variants, introduce full end-to-end watermarking pipelines, often including new decoding rules, custom sampling heads, or learned semantic constraints. In contrast, we present SimKey as a key module approach in of itself, not a full watermarking scheme, as it determines *which* key governs token sampling, and can therefore be paired with many existing mark modules (e.g. ExpMin, SynthID, WaterMax, etc.) without meaningfully altering their sampling distributions.
>
> Because of this modular role, SimKey is hard to compare directly to semantic watermarking systems. SimKey does not impose semantic constraints on the token distribution itself; rather, it ensures that the key selection remains stable under meaning-preserving changes. So, in principle, one could combine SimKey *with* a semantic mark module to improve its internal keying robustness, as we demonstrate with non-semantic mark modules. We have clarified this distinction in the revision.
>
> > **W2: *“The method has hyperparameters (number of keys, number of bits)...study how they affect performance.”***
>
> We appreciate this suggestion and have added a complete hyperparameter ablation. Below we report **TPR at 1% FPR** for a grid of key-index counts (k) and SimHash bit-widths (b). We have included these same results as a heatmap in the revised paper, but need to report the text raw tables here:
>
> **ExpMin**
>
> | k \ b|2   |4   |6   |8    |10   |
> | ---|--|--|--| ---|--|
> | 2    |0.78|0.62|0.57|0.36 |0.35 |
> | 4    |0.51|0.50|0.21|0.36 |0.38 |
> | 6    |0.42|0.34|0.23|0.16 |0.25 |
> | 8    |0.44|0.54|0.17|0.11 |0.19 |
> | 10   |0.29|0.46|0.29|0.075|0.11 |
>
> **WaterMax**
>
> | k \ b|2    |4   |6    |8    |10    |
> | ---| ---|--| ---| ---| ---|
> | 2    |0.70 |0.53|0.45 |0.39 |0.34  |
> | 4    |0.46 |0.30|0.36 |0.087|0.15  |
> | 6    |0.10 |0.12|0.14 |0.087|0.20  |
> | 8    |0.050|0.19|0.050|0.14 |0.062 |
> | 10   |0.075|0.10|0.12 |0.087|0.062 |
>
> From these experiments, several trends emerge. Small (b,k) configurations, such as (b=2, k=2), can produce strong detection signatures but at the cost of noticeably degraded fluency, primarily, as we observed in our experimentation, due to increased key collisions that make sampling more deterministic. As a result, these settings are not practical despite their high TPR. In contrast, the configuration used in the paper, (b=4, k=4), sits on a natural Pareto frontier: it provides consistent robustness gains while keeping perplexity and distributional distortion nearly indistinguishable from standard hashing. Pushing either parameter much larger generally reduced robustness, as the resulting keys become overly sparse and harder to match under semantic perturbations. We have included this hyperparameter study in the revised version of the paper, thank you.

---

> > ### Author Response · Authors · 2025-11-21
> >
> > > **W3: *“...experiments are relatively limited in terms of the number of models and watermark methods considered...”***
> >
> > Thank you for encouraging us to broaden our evaluation. We have done so along three axes, describing each below.
> >
> > **(A) Longer generations**
> >
> > We now include experiments on **500-token outputs**, using the same perturbation types as in the main paper (both semantically related and unrelated substitutions). SimKey continues to provide substantial robustness improvements in all mark modules:
> >
> > | Method  | Unrel-50tok St.Hash | Unrel-50tok **SimKey** | Rel-50tok St.Hash | Rel-50tok **SimKey** |
> > |--|--|--|--|--|
> > | ExpMin  | 0.638 | **0.800** | 0.113 | **0.888** |
> > | SynthID | 0.800 | **0.813** | 0.888 | **0.925** |
> > | WaterMax | 0.400 | **0.563** | 0.138 | **0.788** |
> >
> >
> > This shows that SimKey’s semantic stability scales to much longer contexts and remains effective even when semantic drift accumulates across many sentences.
> >
> >
> > **(B) Alternative embedding models**
> >
> > To examine dependence on the semantic encoder, we replaced *all-MiniLM-L6-v2* with *BAAI/bge-small-en* and repeated our robustness tests. The relative advantage of SimKey persists:
> >
> > | Method  | Unrel-10tok St.Hash | Unrel-10tok **SimKey**|Rel-10tok St.Hash|Rel-10tok **SimKey** |
> > |--|--|--|--|--|
> > | ExpMin  |0.500 |**0.863**|0.250|**0.950** |
> > | SynthID |0.950 |**0.975**|0.825|**0.988** |
> > | WaterMax|0.450 |**0.813**|0.113|**0.800** |
> >
> > These results suggest that SimKey’s improvements do not depend on a particular embedding model.
> >
> > **(C) Multilingual robustness**
> >
> > We also evaluated a **French-language** generation and perturbation pipeline. Mirroring the English results, SimKey improves robustness in nearly all conditions:
> >
> > | Method  |Unrel-10tok St.Hash|Unrel-10tok **SimKey**|Rel-10tok St.Hash|Rel-10tok **SimKey** |
> > |--|--|--|--|--|
> > | ExpMin  |0.475|**0.788**|0.250 |**0.713**  |
> > | SynthID |0.975|**0.988**|0.713|**0.963**  |
> > | WaterMax|**0.463**|0.313|0.138|**0.338** |
> >
> > We believe these complementary experiments broaden the empirical picture. SimKey’s semantic stability is not restricted to a particular language or model configuration. We have included all of these results in the revised version of the paper, and thank you for your suggestion.

---

> > > ### Author Response · Authors · 2025-11-25
> > >
> > > Thank you very much, once again, for your excellent comments. **We are happy to continue the discussion any time until the end of the rebuttal period.** Thank you!

---

> > > > ### Comment · Reviewer_3XpJ · 2025-11-26
> > > > **Response to Rebuttal**
> > > >
> > > > I appreciate the authors for the detailed response and additional experiments. The discussion on the hyperparameter is beneficial. I increased my score to 6. While I agree with the authors that the proposed method is a key module, it might be hard to compare directly to semantic watermarking systems. I still think it would be interesting for the community to see if the proposed key module, combined with the existing mark module, can outperform existing semantic watermarking methods. If so, it could motivate more modular investigation of watermarking methods. But I do understand it could be hard to include additional baselines given the limited time of the rebuttal.

---

> > > > > ### Author Response · Authors · 2025-11-26
> > > > >
> > > > > We thank you for the score raise, and appreciate this added perspective! We agree that a modular investigation of watermarking methods is a promising meta-approach for the community to consider, as iterative stand-alone improvements to key and mark modules continue. We will add a brief discussion of this as a framing for future work in the final revised version of our paper. Again, thank you for your time and consideration, and for helping us make the paper stronger.

---

### Official Review · Reviewer_eTKo · 2025-11-01

**Soundness:** 3
**Presentation:** 2
**Contribution:** 3
**Rating:** 4
**Confidence:** 4

**Summary:**

This paper introduces SIMKEY, a new semantic key module designed to improve the robustness and safety of LLM watermarking. Unlike conventional watermarking methods that generate cryptographic keys based on token-level sequences, SIMKEY ties key generation to the semantic meaning of the context using locality-sensitive hashing over semantic embeddings.

**Strengths:**

1.	The paper identifies a fundamental limitation in existing watermarking schemes such as brittleness to paraphrasing and false attribution. Based on these limitations, the authors propose a semantic keying mechanism based on SimHash. This direction is both theoretically sound and practically impactful.

2.	SIMKEY acts as a plug-and-play module compatible with multiple watermarking schemes. This modularity broadens applicability and promotes reproducibility.

3.	Evaluation across three mark modules, multiple perturbation types, and diverse models offers convincing evidence of effectiveness. Results show consistent TPR improvements under semantic perturbations while preserving fluency.

**Weaknesses:**

1.	Experiments are limited to short generations (prompt + sentence-length texts). It remains unclear how SIMKEY behaves for long-context generation or conversational scenarios where semantics shift gradually.

2.	While SimHash theoretically preserves similarity, the paper lacks an empirical study of false positive rates (FPR) caused by semantically similar but contextually distinct passages.

3.	The use of a fixed semantic encoder (MiniLM-L6-v2) introduces a scenario where performance might vary substantially with different embedders or multilingual contexts.

**Questions:**

-	How does SIMKEY perform under adversarial paraphrasing optimized via gradient-based or LLM-in-the-loop attacks, beyond random translation/substitution?

-	Could the semantic embedding introduce leakage or privacy risks when used in distributed watermark detection pipelines?

-	How sensitive is robustness to the number of hash bits (b) and key indices (k)? Is there an optimal trade-off curve?

---

> ### Author Response · Authors · 2025-11-21
>
> We thank the reviewer for their time and their constructive feedback. We appreciate that they found our paper both “theoretically sound and practically impactful,” highlighting our modularity and reproducibility as strengths, and noted the breadth of our evaluation across multiple watermarking schemes and perturbation settings.
>
> > **W1: “Experiments are limited to short generations (prompt + sentence-length texts)...how SIMKEY behaves for long-context generation or conversational scenarios”**
>
> Thank you for raising this point; we agree that behavior in longer contexts and conversational settings is important for practical deployment.
>
> We have now run **long-context experiments (500-token generations)** using the same attack settings as Table 1, with 50 token-level perturbations per sample. The results below (TPR @ 1% FPR) follow the same trend as in the main paper:
>
> | Method  |Unrel-50tok St.Hash|Unrel-50tok **SimKey**|Rel-50tok St.Hash|Rel-50tok **SimKey** |
> |--|--|--|--|--|
> | ExpMin  |0.638 |**0.800**|0.113|**0.888** |
> | SynthID |0.800|**0.813**|0.888|**0.925** |
> | WaterMax|0.400 |**0.563**|0.138|**0.788** |
>
> The results are consistent with those reported in the main paper: **SimKey improves true positive rate under both unrelated and related semantic perturbations across all three mark modules.**
>
> We also have results for a **conversational test**, where we watermark model responses in a  multi-turn dialogue to evaluate SimKey under gradual semantic drift.
>
> In this setup, an *unwatermarked* LLM (GPT-5.1 inference) produces a scripted sequence of three prompts forming a coherent conversation of roughly 150-200 words, while the **SimKey-watermarked model** (using ExpMin for speed) generates short replies of 10-30 words. We chose topics designed so that semantics shift relatively naturally from turn to turn. For example, one set of dialogue questions began with *“What running shoes help with mild overpronation?”* and end with *“Could you outline a simple 5K plan?”*, while another moved from *“How often should I water a pothos?”* to *“How can I design a low-light plant shelf?”*.
>
> Only the model’s responses are watermarked; all user prompts are fixed and unwatermarked. We ran 20 such conversations across diverse topics, watermarking with SimKey+the ExpMin mark module. We observed reliable watermark detection, even at a low FPR: **TPR @ 1% FPR = 0.800**. We have included a description of this experiment and the result in our revision.
>
> > **W2: “the paper lacks an empirical study of false positive rates (FPR) caused by semantically similar but contextually distinct passages.”**
>
> We appreciate this clarification request and agree that unintended attribution to semantically similar but contextually distinct text is an important failure mode to examine. Before running additional experiments, **we would like to confirm that we are interpreting your concern correctly.**
>
> The experiment we have in mind is the following: we would construct clusters of **semantically similar but contextually distinct passages**. For example, different sports articles or multiple news reports describing unrelated events, using the same embedding model.
>
> Within each cluster, we would watermark only a single passage, and then run detection on **all** passages in the cluster, measuring the false positive rate on the non-watermarked yet semantically similar texts. We would compare SimKey and standard hashing under equal TPR to determine whether SimKey increases the chance of accidental attribution in this setting. This would check whether any amount or increase of false positives arises from semantic similarity alone. Please let us know if this is what you had in mind, and we can run this experiment and report our results, and include it in the final revision, thank you.
>
> > **W3: “performance might vary substantially with different embedders or multilingual contexts.”**
>
> We appreciate this point, and now have included results on an alternate embedding models to show SimKey’s robustness to this design choice.
>
> Instead of *all-MiniLM-L6-v2*, we use *BAAI/bge-small-en* as the semantic encoder and repeat the token replacement robustness experiments (TPR @ 1% FPR):
>
> | Method  |Unrel-10tok St.Hash|Unrel-10tok **SimKey**|Rel-10tok St.Hash|Rel-10tok **SimKey** |
> |--|--|--|--|--|
> | ExpMin  |0.500 |**0.863**|0.250|**0.950** |
> | SynthID |0.950|**0.975**|0.825|**0.988** |
> | WaterMax|0.450|**0.813**|0.113|**0.800**|
>
> The relative pattern is very similar to what we observe with MiniLM, where SimKey consistently improves robustness under both unrelated and related edits across mark modules.

---

> > ### Author Response · Authors · 2025-11-21
> >
> > To demonstrate multi-lingual robustness, we also tested **French** text using the same pipeline (generation, unrelated and related token replacements), again reporting TPR @ 1% FPR:
> > | Method  |Unrel-10tok St.Hash|Unrel-10tok **SimKey**|Rel-10tok St.Hash|Rel-10tok **SimKey** |
> > |--|--|--|--|--|
> > | ExpMin  |0.475|**0.788**|0.250 |**0.713**  |
> > | SynthID |0.975|**0.988**|0.713|**0.963**  |
> > | WaterMax|**0.463**|0.313|0.138|**0.338** |
> >
> > Again, SimKey generally improves robustness, including in the non-English setting. These results suggest that **SimKey’s benefits are not specific to a particular encoder or strictly to English text**. We have incorporated these tables and a short discussion of the experiments in the paper revision, thank you again for the suggestion.
> >
> > > **Q1: “How does SIMKEY perform under adversarial paraphrasing optimized via gradient-based or LLM-in-the-loop attacks”**
> >
> > We agree that e.g. LLM-in-the-loop paraphrasing is an important and realistic attack model. Following this suggestion, we ran an **LLM-based paraphrasing attack** inspired by the setup in *Watermarks in the Sand* (Zhang et al., 2024).
> >
> > The attack protocol we used is as follows: (1) we first generate watermarked text using the base LLM with e.g. ExpMin as the mark module and with either standard hashing or SimKey as the key module; (2) we then apply an LLM paraphraser twice in succession, producing a paraphrase-of-a-paraphrase, using a prompt instructing the model to rewrite the text while preserving meaning exactly and providing no commentary; and (3) we run watermark detection on the final paraphrased output and compute the **TPR at 1% FPR**.
> >
> > To better understand the semantic drift introduced by this attack, we also measured BERTScore similarity across paraphrasing steps. Across our sample of 80 paraphrases, the first paraphrase has a mean similarity of 0.7105 (std 0.1000) with the original phrase, and the second paraphrase has a mean similarity of 0.6868 (std 0.1128). This suggests that the paraphraser introduced nontrivial semantic drift even after one rewrite, and that the two-step paraphrasing compounds this drift.
> >
> > Under this attack, we obtain the following detection rates: ExpMin + **SimKey** achieves 0.125 versus 0.0625 for ExpMin + standard hashing, and WaterMax + **SimKey** achieves 0.0875 versus 0.0125 for WaterMax + standard hashing.
> >
> > Perhaps as expected, strong paraphrasing significantly reduces detectability for all methods, but **SimKey consistently yields noticeably higher detection rates than standard hashing under the same low FPR**, indicating that semantic keying preserves more recoverable signal even under aggressive LLM-based paraphrasing. We have included this adversarial paraphrasing attack in the revised paper, thank you for suggesting it.
> >
> > > **Q2: “Could the semantic embedding introduce leakage or privacy risks when used in distributed watermark detection pipelines?”**
> >
> > This is an important point for clarification, thank you for raising it! SimKey **does not transmit or expose semantic embeddings during generation or detection**. The embedding is computed locally from already-visible context tokens, the same as in a non-watermarked text generation. No new information channel is introduced beyond standard autoregressive access. We have clarified this in the revision.

---

> > > ### Author Response · Authors · 2025-11-21
> > >
> > > > **Q3: “How sensitive is robustness to the number of hash bits (b) and key indices (k)?”**
> > >
> > > We appreciate this question. To study this, we ran a **2D ablation** over the number of hash bits (b) and key indices (k), varying one along rows and the other along columns, and reporting TPR @ 1% FPR. Below we show results for ExpMin and WaterMax (SimKey as key module):
> > >
> > > ExpMin
> > > | k \ b|2    |4    |6    |8     |10    |
> > > |-----|-----|-----|-----|----|-----|
> > > | 2    |0.78 |0.62 |0.57 |0.36  |0.35  |
> > > | 4    |0.51 |0.50 |0.21 |0.36  |0.38  |
> > > | 6    |0.42 |0.34 |0.23 |0.16  |0.25  |
> > > | 8    |0.44 |0.54 |0.17 |0.11  |0.19  |
> > > | 10   |0.29 |0.46 |0.29 |0.075 |0.11  |
> > >
> > > WaterMax
> > > | k \ b|2    |4    |6    |8    |10    |
> > > |-----|-----|-----|-----|-----|-----|
> > > | 2    |0.70 |0.53 |0.45 |0.39 |0.34  |
> > > | 4    |0.46 |0.30 |0.36 |0.087|0.15  |
> > > | 6    |0.10 |0.12 |0.14 |0.087|0.20  |
> > > | 8    |0.050|0.19 |0.050|0.14 |0.062 |
> > > | 10   |0.075|0.10 |0.12 |0.087|0.062 |
> > >
> > > We observe the following pattern in the ablation grid. A very small configurations, such as (b=2, k=2), produces strong correlations and a high TPR, but we found these also lead to **noticeable degradation in text quality**, driven by increased key collisions and more deterministic sampling; as a result, these settings are not desirable in practice.
> > >
> > > By contrast, the configuration we use in the paper, **(b=4, k=4)**, sits on a natural **Pareto frontier between robustness and fluency**: we found that it yielded a substantial TPR improvement while keeping perplexity nearly identical to standard hashing. We found that pushing either (b) or (k) much higher harmed robustness, as in the keys become overly sparse or varied, while reducing the chance of matching under perturbations, and without offering compensating gains in quality. We have included these results as a heatmap in the revision and summarized these trade-offs.

---

> > > > ### Author Response · Authors · 2025-11-25
> > > >
> > > > Thank you very much again for your insightful comments. **We’re happy to continue the conversation at any point throughout the discussion period.** Thanks!

---

### Author Response · Authors · 2025-11-23
**General Response**

We appreciate the time and effort that all the reviewers gave to our paper, and thank them for their valuable feedback. Our work introduces SimKey, a semantic keying module for language models watermarks. We appreciate that the reviewers find our technique effective (*eTKo*, *3XpJ*, *79EQ*), the problem we study impactful (*eTKo*, *Q2FX*, *79EQ*), and our keying technique widely applicable (*eTKo*, *3XpJ*, *79EQ*).

Following your suggestions, we highlight further improvements:
1. Evaluations using **longer generation, including multi-turn conversations.** SimKey consistently boosts detection across all three mark modules.
2. Evaluations against **additional LLM-in-the-loop removal attacks** (via paraphrasing). While paraphrasing significantly reduces detectability for all methods, SimKey consistently yields noticeably higher detection rates
3. Evaluations of our methods in additional settings: including **ablations of our $k$ and $b$ parameters, additional semantic embedders, and an additional language.** Our results remain consistent with those originally reported.

Our revised submission incorporates these improvements along with our responses to the other reviewers' comments, where changes have been highlighted in blue.

**We would be very happy to keep the discussion going,** addressing any points that remain unclear, or any new suggestions. Thanks again for your suggestions!

---

### Meta-Review · Area_Chair_xUTo · 2026-01-08

**Summary:**

This paper proposes SIMKEY, a semantic key module based on semantic hashing to improve the robustness of LLM watermarking. SIMKEY generates keys that are tied to the LSH hashes of the semantic meaning of the context. SIMKEY can also be paired with many different mark modules. SIMKEY is empirically shown to improve robustness against various attacks while preserving generation perplexity when using as the key module in multiple watermarking methods.

The reviewers have the following main and important concerns: SIMKEY’s behavior under adversarial attacks such as paraphrasing/spoofing attacks, limited number of models/watermarking methods studied, privacy leakage, and hyperparameter sensitivity study. The authors have provided responses to all concerns, and issues such as privacy and hyperparameter sensitivity, along with other clarifications and minor comments, are well addressed. Nevertheless, the paper’s evaluation is still limited to 2 LLMs within the same family of models. In addition, additional analysis on the failures under paraphrasing should be conducted (e.g., it’s unclear why SIMKEY has better performance under paraphrasing, what is its performance behavior against paraphrasing on longer text or against simpler methods such as synonym substitution). I believe the paper still lacks rigorous evaluation and insights for a publication.

**Reviewer Concerns:**

- eTKo
   - Experiments are limited to short generations (prompt + sentence-length texts).
   - the paper lacks an empirical study of false positive rates (FPR) caused by semantically similar but contextually distinct passages.
   - The use of a fixed semantic encoder (MiniLM-L6-v2) introduces a scenario where performance might vary substantially with different embedders or multilingual contexts.
   - How does SIMKEY perform under adversarial paraphrasing optimized via gradient-based or LLM-in-the-loop attacks, beyond random translation/substitution?
   - Could the semantic embedding introduce leakage or privacy risks when used in distributed watermark detection pipelines?
   - How sensitive is robustness to the number of hash bits (b) and key indices (k)? Is there an optimal trade-off curve?
- 3XpJ
   - Semantic Watermarking methods in the Section 6, there are no comparisons
   - How hyper-parameter configuration affect the performance of the proposed method is not well studied.
   - For a pure empirical paper, the experiments are relatively limited in terms of number of models and number of watermark methods considered.
- Q2FX
   - Further details and clarification on the effective # of keys for a certain topic? How does the effective # of keys affect detection?
   - more adversarial thinking on how SimKey can be circumvented.
- 79EQ
   - lacks a theoretical justification for its claimed distortion-free property
   - The reported improvements in attack resistance are relatively small and inconsistent
   - The evaluation setup is insufficiently described
   - The paper does not address potential countermeasures against spoofing attacks, in which adversaries generate fake watermarks to mislead detection

The authors have responded to all concerns in the rebuttal. However, I believe the results on paraphrasing require further analysis, while the paper is still limited to 1 family of text-generation LLMs.

**Reviewer Scores:**

- eTKo: 4 - unclear, this reviewer likely has more questions toward the authors, specifically on paraphrasing results
- 3XpJ: 4 - increase score to 6
- Q2FXL: 8 - keep score, not an expert in watermarking
- 79EQ: 4 - unclear, this review likely still keeps his score due to the limited number of evaluated LLMs

---

### Decision · Program_Chairs · 2026-01-26

Reject